# Learnability of the Boolean Innerproduct in Deep Neural Networks

**DOI:** 10.3390/e24081117

**Published:** 2022-08-13

**Authors:** Mehmet Erdal, Friedhelm Schwenker

**Affiliations:** Institute of Neural Information Processing, Ulm University, 89081 Ulm, Germany

**Keywords:** deep learning, circuit complexity, neural networks

## Abstract

In this paper, we study the learnability of the Boolean inner product by a systematic simulation study. The family of the Boolean inner product function is known to be representable by neural networks of threshold neurons of depth 3 with only 2n+1 units (*n* the input dimension)—whereas an exact representation by a depth 2 network cannot possibly be of polynomial size. This result can be seen as a strong argument for deep neural network architectures. In our study, we found that this depth 3 architecture of the Boolean inner product is difficult to train, much harder than the depth 2 network, at least for the small input size scenarios n≤16. Nonetheless, the accuracy of the deep architecture increased with the dimension of the input space to 94% on average, which means that multiple restarts are needed to find the compact depth 3 architecture. Replacing the fully connected first layer by a partially connected layer (a kind of convolutional layer sparsely connected with weight sharing) can significantly improve the learning performance up to 99% accuracy in simulations. Another way to improve the learnability of the compact depth 3 representation of the inner product could be achieved by adding just a few additional units into the first hidden layer.

## 1. Introduction

Deep learning models have revolutionized the field of machine learning by achieving state-of-the-art results in the hardest challenges such as speech recognition, the recognition and detection of visual objects or the classification of complex spatio-temporal patterns [1,2,3]. Even so, little has been understood about how they work in detail. One possible explanation for the success of deep learning models could be that they are able to combine feature extraction at several levels of abstraction and classification into one single model while keeping its size at a computationally feasible level. The size of an architecture is basically defined by the number of computational units and it is strongly assumed that deep architectures need significantly less of them than their shallow counterparts [4,5]. However, how many hidden layers and units per hidden layer are needed to solve a particular task is generally unknown [6]. After all, the assumption that a deep architecture can compactly represent a highly nonlinear function is based on findings in the research field of circuit complexity, where it was observed that the size of linear threshold networks of some Boolean functions can be exponentially reduced by increasing the depth, i.e., the number of layers [7,8,9]. Since then, several attempts have been made to show that this representational advantage of increasing depth also applies to other function spaces that are more relevant to practical applications and would thus explain the success of deep learning models [10,11,12,13]. However, although it is necessary that a model can represent the function to be learned, it is not sufficient to explain the success of deep models. Even the most efficient representation of a function does not automatically imply learnability [14]. Rather, it is a combination of the architecture, learning method, and quality of available training data on which learnability depends [15,16].

### 1.1. Related Work

Since, for certain Boolean functions, somehow exactly fitting architectures are a priori known and the whole noise-free sample space is available as training data, Boolean functions are often used to study the impact of deep architectures in supervised learning using gradient descent-based methods [17,18,19,20,21,22,23,24,25]. Different aspects have been considered in the literature. For example, in [19], the learnability of the Boolean parity function with an hand-coded deep architecture with input size d={16,32,64,128} was investigated using a randomly selected subset of the Boolean inputs for training. The generalization performance was then measured on an unseen test set. They concluded that the parity function is basically not learnable with gradient-based methods based on subsets as training material. In fact, the parity function is one of the hardest nonlinear problems when using as few computational units as possible and backpropagation as the learning algorithm [26,27]. Moreover, the Boolean inner product can be used to investigate whether a neural network type is generally suitable for certain tasks. As such, in [28], it is shown that Restricted Boltzmann Machines (RBM) cannot learn the Boolean inner product and therefore more layers might be needed to learn the unknown probability distribution of a given training set. This could explain why deeper generative adversarial networks and variational autoencoders are more successful [29].

### 1.2. Learning of the Boolean Inner Product

In other works, the learnability of parity or a Boolean inner product with certain architectures is usually evaluated based on the generalization performance. In this work, we investigated the learnability of the Boolean inner product by using the full input dataset as training data. Thus, we consider learning from the point of view of optimization rather than generalization. In our study, we focus on the smaller dimensions d={10,12,14,16}. Our goal is to better understand the extent to which the choice of architecture affects supervised learning with stochastic gradient descent (SGD). The Boolean inner product is particularly well suited for this purpose since it can be represented by an exponentially smaller architecture when another hidden layer is added [7,9]. Since not only the complete input space of the Boolean inner product is known, but also the relevant patterns in the input data, we further investigate the extent to which the learning performance can be improved if the first hidden layer is adapted to the input patterns with partial connections.

## 2. Materials and Methods

In deep learning, an unknown function is usually to be learned using only a small fraction of possible examples as training data. In that case, the goal is rather to obtain a function that correctly classifies examples produced by the same process as the training data. This capability is referred to as *generalization*, and the achievable performance depends not only on the network architecture but largely on the quality and quantity of the training data [30,31,32]. In this work, we were not concerned with generalization, but with learning a particular Boolean function $t:{\left\{0,1\right\}}^{n}\mapsto \left\{0,1\right\}$ that is a priori known. The training data consisted of the whole noise-free input space as described in more detail in Section 2.5. As a learning algorithm, the state-of-the-art stochastic gradient descent (SGD) with Nestorov momentum and an adaptive learning rate was used. We will first give a detailed description and some background information about the architectures we studied in this work, and then describe how we used SGD to train them.

### 2.1. Circuit Complexity of the Boolean Inner Product

Circuit complexity plays a special role in the history of deep learning. The central question that connects both areas is how many layers can help to efficiently model a function in terms of the number of units. We call a circuit or a neural network a depth-*k* network if it has k−1 hidden layers and one output layer. It is well known that any Boolean function can be exactly represented by a depth-2 network of threshold neurons [33]. These networks are typically based on the disjunctive normal form (DNF) and can grow exponentially with the network’s input size *n*. One of these functions with the exponentially growing DNF network is the *n*-digit parity function $P_{n}:{\left\{0,1\right\}}^{n}\mapsto \left\{0,1\right\}$. Parity is a symmetric Boolean function and can thus be represented by depth-2 networks with at most n+1 units. In the theory of circuit complexity, the set of parity functions $P=\{P_{n}|n\in N\}$ belongs to the complexity class ${\mathrm{TC}}^{0}$ containing the Boolean functions that can be represented by a circuit of constant depth and polynomial size [8]. Let ${\mathrm{TC}}_{k}^{0}$ be the set of functions with a depth-*k* architecture, then ${\mathrm{TC}}^{0}={⋃}_{k=0}^{\infty }{\mathrm{TC}}_{k}^{0}$ with the hierarchical relationship
(1)$${\mathrm{TC}}_{0}^{0}\subseteq {\mathrm{TC}}_{1}^{0}\subseteq {\mathrm{TC}}_{2}^{0}\subseteq {\mathrm{TC}}_{3}^{0}\ldots$$

While P is in ${\mathrm{TC}}_{2}^{0}$, there is an interesting open research question regarding this hierarchy [9]: is there a network in ${\mathrm{TC}}_{3}^{0}$ which can represent all the functions from ${\mathrm{TC}}^{0}$? This question is related to the Boolean inner product $I_{n}:{\left\{0,1\right\}}^{2n}\mapsto \left\{0,1\right\}$, since for this function, it is known that a depth-3 network is necessary to be in ${\mathrm{TC}}^{0}$. This means that $I=\{I_{n}|n\in N\}$ is in ${\mathrm{TC}}_{3}^{0}$, because a depth-2 network needs exponentially more units than a depth-3 network to represent $I_{n}$. Thus, at least a network of depth 3 is necessary to represent all functions of ${\mathrm{TC}}^{0}$, but we do not yet know whether further layers really help or if the hierarchy collapses at ${\mathrm{TC}}_{3}^{0}$. A similar question can also be raised with respect to the learnability of $I_{n}$: is $I_{n}$ as a depth-3 network efficiently learnable or are more hidden layers needed? In this work, we empirically demonstrate that $I_{n}$ seems to be efficiently learnable with a depth-3 network if the fully connected input layer is replaced by a convolutional layer as input layer.

### 2.2. Network Architectures

The architecture of a neural network is usually designed by trial and error, since there are no proven rules to find an optimal architecture for a particular task [34]. Nevertheless, it is at least known that any continuous function defined on a compact subset of $R^{n}$ can be approximated with any desired accuracy by a neural network with only one hidden layer [35]. However, exactly how many units are needed to approximate a function implicitly given by a set of examples with a depth-2 network is generally unknown. Fortunately, in the Boolean space, at least the size of the linear threshold network is known, which represents the disjunctive normal form. In the following sections, we present both the linear threshold networks of the parity function and its extension, i.e., the Boolean inner product whose learnability we want to investigate. The only activity function used in a linear threshold network is the linear threshold function *f*, which is defined as follows: (2)$$f_{w}\left(x\right)=g\left(\sum_{i=1}^{n}x_{i}w_{i}-\Theta \right)\;\mathrm{with}\;g\left(y\right)=\{\begin{array}{cc} 1 & y\geq 0\\ 0 & else \end{array}$$
where $x=(x_{1},\ldots ,x_{n})$ is the input and $w=(w_{1},\ldots ,w_{n},\Theta )$ is the state of the unit. Later, in the simulations, the linear threshold units are replaced by a smoother activation function. All fully connected architectures described here can be found in [9].

### 2.3. Parity Function

Since the *n*-digit parity function can be defined with the XOR-operator only, it can be seen as a kind of escalation of the XOR-problem. To illustrate the hardness of this problem, Figure 1 shows how the distribution of positive and negative examples within the corresponding hypercube becomes progressively worse from n=2 to 4.

While the parity problem (n=2) can be solved by a small feed forward neural network, it is so far unclear whether the parity function for a large *n* can be learned with neural networks at all. Anyway, the parity function $P_{n}:{\left\{0,1\right\}}^{n}\mapsto \left\{0,1\right\}$ is given by:(3)$$P_{n}\left(x\right)=\left(\sum_{i=0}^{n}x_{i}\right)\;\mathrm{mod}\;2$$

It is $P_{n}\left(x\right)=1$ if $x=(x_{0},\ldots ,x_{n-1})$ contains an odd number of ones. This is the case for exactly $2^{n-1}$ inputs. Any Boolean function ${\left\{0,1\right\}}^{n}\mapsto \left\{0,1\right\}$ can be represented in disjunctive normal form (DNF). Furthermore, any Boolean expression in DNF can be computed by a neural network with one hidden layer where, for each positive example $x^{\left(j\right)}=\left(x_{0},\ldots ,x_{n-1}\right)$, there is a hidden threshold unit $h_{w_{j}}$ representing the associated conjunction of the DNF with $w_{j}=\left(w_{1},\ldots ,w_{n},\Theta_{j}\right)$ whereas threshold $\Theta_{j}$ is the number of ones in $x^{\left(j\right)}$ and the weights $w_{1},\ldots ,w_{n}$ have the values: (4)$$w_{i}=\{\begin{array}{cc} 1 & x_{i}=1\\ -1 & else \end{array}$$ Finally, the output layer consists of a single threshold unit computing the disjunction of the outputs received from the single hidden layer. Because a DNF formula is true if at least one conjunction is true, the threshold function $g_{w}$ with $w=\left(w_{1},\ldots ,w_{n},\Theta_{j}\right)=\left(1,1,\ldots ,1\right)$ can be used as the output unit. After all, the size of the architecture for the DNF requires $2^{n-1}+1$ threshold units and thus exponentially grows with the input dimension *n*.

Figure 2 shows a DNF architecture for n=4 and even for that small architecture, the number of hidden units is already twice as large as the input dimension. Apparently, with an increasing *n*, this architecture becomes very fast, and a universal approximator capable of representing all functions defined on the corresponding input space. Such an architecture can memorize the whole input space and one could expect that the learning problem is thus relatively easy, at least in theory. However, the main problem is that both the input space and the number of hidden units grows exponentially with *n* and finding a solution by searching the whole hypothesis space becomes practically infeasible.

#### Exact Architecture

Due the symmetric nature of the parity function, the number of required units can be drastically reduced to n+1. In this reduced architecture, the hidden units $h_{w_{1}},\ldots ,h_{w_{n}}$ count the number of ones of the input with $w_{j}=\left(w_{1},\ldots ,w_{n},\Theta_{j}\right)=\left(1,1,\ldots ,1,j\right)$. Now, the output unit $g_{w}$ needs to classify the hidden output $\left(h_{w_{1}}\left(x\right),\ldots ,h_{w_{n}}\left(x\right)\right)$ with an odd number of ones as class 1 which can be achieved using the state $w=(w_{1},\ldots ,w_{n},\Theta_{j})$ with $w_{i}={\left(-1\right)}^{i+1}$. Because this architecture is a kind of fitting exactly solution, we call it exact-architecture. Figure 3 shows an example of the exact-architecture for n=4. This architecture grows linearly with the input dimension *n* whereas the hidden layer has always the same size as the input layer. Assuming that in any case the entire input space is needed to learn the function, the problem is still infeasible for large *n*. However, the exact-architecture should converge faster and the infeasible *n* should be notably larger than for the DNF architecture. Nonetheless, the question arises if the learning problem of parity has become even harder now, because that kind of minimal architecture has significantly fewer parameters and is consequently far less fault tolerant.

### 2.4. Boolean Innerproduct

The Boolean Innerproduct $I_{n}:{\left\{0,1\right\}}^{n}\times {\left\{0,1\right\}}^{n}\mapsto \left\{0,1\right\}$ is defined as follows:(5)$$I_{n}\left(x,y\right)=\left(\sum_{i=1}^{n}x_{i}y_{i}\right)\;\mathrm{mod}\;2$$

It can be thought as an extension of the parity problem, since before calculating the parity, the matching ones $\left(x_{i}=1,y_{i}=1\right)$ in the two Boolean vectors x and y are determined using their inner product. Consequently, it is $I_{n}\left(x,\;y\right)=1$ if the example $\left(x_{1},\ldots ,x_{n},y_{1},\ldots ,y_{n}\right)$ contains an odd number of matching ones $x_{i}=y_{i}=1$. Since there are $2^{2n-1}-2^{n-1}$ positive examples (a proof of the number of positive examples can be found in the Appendix A), the corresponding shallow DNF architecture grows exponentially with the input dimension *n* as well. However, to obtain an architecture of linear size, an additional hidden layer is needed now. The first hidden layer computes the Boolean AND for each $\left(x_{i},\;y_{i}\right)$ using threshold units $f_{w_{1}},\ldots f_{w_{n}}$ of the form shown in Figure 4.

Most of the weights of the first hidden layer are 0, however, the rest of the network computes the parity function with the exact architecture described in Section 2.3. This results in a deep exact architecture with overall two hidden layers. In the example for n=3 in Figure 5, the 0-edges in the first hidden layer are drawn with dotted lines to indicate that they do not contribute anything to the computation. Nonetheless, these connections must also be learned, which could be very difficult since all weights are equally updated when using backpropagation as learning algorithm. Unfortunately, there is not really another solution, because the first hidden layer must map the input to a dual number representing the number of matching ones. Thus, this can only be performed by a two-digit Boolean operator that assigns a different value to each matching one $x_{i}=y_{i}=1$ than to the other combinations. The only other function with this property is the NAND operator whose representation as a threshold unit has just the negated version $-w_{i}$ of the weights of the AND threshold unit.

#### Partially Connected First Hidden Layer

The learning problem can be simplified, if all 0-edges are removed from the first hidden layer. Furthermore, since all threshold units use the same weights to calculate the AND function, only one configuration $\left(w_{1},w_{2},\Theta \right)$ is needed which can be shared across the whole layer. In other words, the same pattern $x_{i}=1,y_{i}=1$ occurs at different locations in the input data and thus only one single translation invariant filter is needed. This kind of problem is common in image recognition where the location of the same pattern can vary between different images. To address this problem, the so-called convolutional layer is used [36]. For the Boolean inner product, a 1D convolutional layer as a first hidden layer with one kernel of the form $\left(w_{1},0,\ldots ,0,w_{\left(n/2+1\right)}\right)$ would be sufficient. However, the kernel can actually be reduced to $\left(w_{1},w_{2}\right)$ if the input is first reshaped into a (n×2) matrix. The (1×2)-filter kernel is then shifted row-wise over the input as demonstrated in Figure 6.

The learning problem can be made harder again by adding additional rows to the filter whereas by using an (n×2)-filter, a similar performance as with the fully connected layer should be achieved.

### 2.5. Simulation Setup

All simulations are implemented and performed with the software library TensorFlow [37]. As the optimization algorithm, mini-batch gradient descent with an adaptive learning rate (lr=0.3, $decay=10^{-8}$) and Nesterov momentum (m=0.9375) [38] was used. For the all depth-3 architecture, the learning rate had to be reduced to lr=0.03. To approximate the threshold function of the linear threshold unit network solution as accurately as possible, the sigmoid is used as an activation function in all layers of all networks (we also tried the so-called Hard sigmoid activation function, but since it did not perform well, we did not pursue it further). Because all networks are basically binary classifiers, no normalized activation function such as softmax is needed in the output layer. As training data, all possible examples are used. Let $t:{\left\{0,1\right\}}^{n}\mapsto \left\{0,1\right\}$ be the function to be learned, the training data are then defined as $D=\{\left(x,t\left(x\right)\right)|x\in {\left\{0,1\right\}}^{n}\}$. Since the output t(x)∈{0,1} is always binary, we used binary cross-entropy as the loss function. In all simulations, the classification accuracy is used as the performance metric, sometimes abbreviated with *acc*. The classification accuracy is defined as the fraction of correct predictions and the size |D| of the training data. To evaluate *n* simulations, the average accuracy is used, which is simply the mean value of all *n* single accuracy values.

## 3. Results

Since the Boolean inner product is based on the parity function, we first looked how parity can be directly learned. The parity function is notorious for being very hard to learn with back-propagation [39,40,41,42], especially when somehow minimal networks are used. We then investigated the learnability of the Boolean inner product with the standard solution architectures, the shallow DNF architecture and the deep exact architecture. Finally, we used the deep partially architecture for learning to see if the performance can be improved by adapting the first hidden layer to input problem.

### 3.1. Parity Function

To study the learnability of the parity function, we want to compare the learning performance of the DNF architecture with the exact architecture. We performed simulations for input dimensions of *n* = 5, 6, …, 9 with 100 trials for each architecture. Figure 7 shows that the DNF architecture generally performed better, with the performance difference significantly increasing as *n* increases. In fact, the average accuracy of DNF architecture is similar for all *n* ranging from 95% to 97%. However, the average accuracy of the exact architecture continuously significantly decreases as n increases. At *n* = 9, the accuracy has already dropped by approximately 10%. However, since the average accuracy is highly sensitive to outliers and can therefore be misleading, we added box plots in Figure 7 to verify the results. Especially, for the exact architecture, there is almost no variation within the sample of 100 trials. The average accuracy is consequently a good estimation and the same performance can be expected when using this architecture. The situation is not much worse for the DNF architecture, so that the average accuracy gives a relatively good estimation here as well. From this, we conclude that parity is much harder to learn with the exact architecture, which may indicate that the only linearly growing size of the exact solution is too small. To address the challenge of learning the input layer even more, we adapted the first hidden layer to the input problem by using convolution.

### 3.2. Boolean Inner Product

We now describe the results of our central theme, and the learnability of the Boolean inner product $I_{n}:{\left\{0,1\right\}}^{n}\times {\left\{0,1\right\}}^{n}\mapsto \left\{0,1\right\}$ with the DNF architecture compared to the deep exact architecture. First, Figure 8 shows the number of weights for both architectures to give a better insight into how enormous the size difference between them grows as *n* increases.

While the DNF architecture grows exponentially $2^{O(n)}$ with
$$2^{2n}\left(n+1\right)-2^{n}\left(n+1\right)+2n+3$$
the deep exact architecture grows only quadratic $O(n^{2})$ with
$$3n^{2}+3n+1$$

Even for n=8, 100 trials with the DNF architecture would take several days, which in a way reflects how fast that architecture becomes practically unusable. We therefore performed simulations for the deep exact architecture for n=5,6,7,8 and for the DNF architecture for n=5,6,7. As it can be seen in Figure 9, similar to the parity function, the DNF architectures also perform significantly better, whereby this time, the function was even learned almost completely for each *n*. Only the learning speed decreased with increasing *n*. The corresponding boxplots also show almost no variance, which illustrates how robust these architectures learn. This high performance is most likely due to the enormous size of the networks. The deep architectures, on the other hand, perform significantly worse again. Surprisingly, this time, as both the performance and the learning speed improve, the *n* becomes larger.

This could be an indication that the deep exact architecture needs a lot of training examples. Nevertheless, the result is far from being even close to being learned. Furthermore, the box plots also show very high instability of the performance distributed over all trials. We assume that the difficulty here is mainly due to the first of the two hidden layers. To investigate this assumption, we added two additional neurons to the first hidden layer to give the network more leeway. The result in Figure 10 confirms our assumption in a way.

The performance of all nets as well as the stability of the performance distributed over all trials have both significantly improved, especially with an increasing *n*. Of course, this alone cannot confirm that the first hidden layer specifically is the main problem. The network could also be too small in general. To show that, indeed, the first hidden layer is the most critical part of the architecture, we investigated for n=7 how the learning performance changes when both hidden layers are extended stepwise by 0,1,…,4 neurons. If *a* and *b* are the number of additional neurons in the first and second hidden layer, this results in architecture sizes of (n+a)+(n+b)+1 with $\left(a,b\right)\in {\left\{0,1,\ldots ,4\right\}}^{2}$. Figure 11 shows the overall result in a 5×5 matrix such that *a* numbers the rows and *b* numbers the columns.

Each element of the matrix contains the average accuracy over 25 trials. The first row of the matrix contains the weakest results of all, with the values rising relatively slightly at first and even falling again from the middle. Adding additional neurons to the second hidden layer could therefore only slightly improve the performance. The situation is quite different for the first column, where the values increase much faster and end with one of the best results of all, namely (a, b) extensions with an improvement of more than 10%. This result clearly indicates that the first hidden layer is the problem area of the network. The deep exact architecture seems to need more leeway here to find the unevenly distributed weights of the solution.

### 3.3. Partially Connected Network

Based on previous results, which indicate that the network has enormous difficulty learning the first hidden layer, we now construct a first layer that takes into account the uneven structure of the solution to be learned. For this purpose, we use a convolutional layer with a filter that contains only two weights, namely those needed for the AND operator. We again performed simulations for n=5,6,7,8, but this time, only 10 trials per architecture in the expectation that the solution will be approximated much better, faster and more stable. This expectation is based on the fact that the first hidden layer is now adapted to the input pattern to be learned which should make learning much easier than with a fully connected layer. Figure 12 right shows that our expectation was pretty well met. All networks perform significantly better than their deep exact counterparts. Like with the other architectures, the performance increases again with the input dimension *n*, which in a way underlines the fact that this problem is very likely to require a lot of sample data. To investigate whether it is possible to be less accurate with the convolutional layer filter, we gradually increased the filter from 1×2 to 4×2 for *n* = 7. Figure 12 left shows that the performance already completely collapses at 3×2 and that there is no longer any question of learning.

## 4. Discussion

Since the Boolean inner product is basically a more complicated version of the parity function, we first took a brief look at the learnability of parity. The poor performance of the exact architecture in our simulations confirms in some way the assumption that this function is very hard to learn with gradient descent-based methods. The reason why parity is considered to be one of the hardest functions to learn might be that very similar inputs have different outputs, i.e., classes. The Hamming distance *d* between points with different parity is just one, d=1. The exact architecture has the smallest known number of computational units needed to represent $P_{n}$ which may be insufficient to capture its complex behavior by learning with backpropagation. This is further underlined by the far better performance of the DNF architecture. As a kind of extension of the parity function, the Boolean inner product seems to inherit this problem somehow. Thus, very similar results are obtained in the simulations with $I_{n}$. Again, the far smaller deep exact architecture performed far worse than the exponentially larger DNF architecture. However, in contrast to the results with the parity functions, this time, the performance increases with the input dimension, suggesting that more data could be helpful for learning $I_{n}$ with the deep exact architecture. Based on the assumption that the exact architectural solution is too small and that learning the input layer is the hardest challenge to successfully learn $I_{n}$, we added a few more units to the input layer to create a little bit more leeway. Furthermore, the performance significantly increased, indicating that this assumption could be correct. To address the challenge of learning the input layer even more, we adapted the first hidden layer to the input problem using convolution. With these partially connected architectures, by far the best results could be achieved and it can be said that the Boolean inner product can thus be learned almost perfectly with a depth-3 architecture.

## 5. Conclusions

In this work, we studied the learnability of the Boolean inner product with the exact solution as a deep architecture, which is exponentially smaller than the corresponding shallow architecture. However, our simulations showed that this architecture is significantly more difficult to learn. Further simulations have shown that the network is not simply too small, but seems to need more leeway in the form of a few extra neurons, mainly in the first hidden layer. Finally, we showed that with a convolutional layer adapted to the input problem, this additional leeway is not necessary. The results show that a more input problem-oriented approach could yield much better results. However, one must keep in mind that we studied a very special problem here. In other problems, the main difficulty could also lie in the *n*-th layer. On the other hand, it is well known that feature extraction is one of the biggest challenges in machine learning. Accordingly, much better results with the partially connected networks are promising. However, further in-depth investigations may be useful to understand the learnability in deep networks.

## Figures and Tables

**Figure 1 entropy-24-01117-f001:**
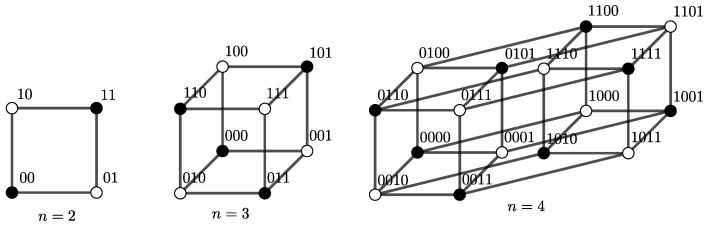
Hypercubes of the parity for n=2,3,4.

**Figure 2 entropy-24-01117-f002:**
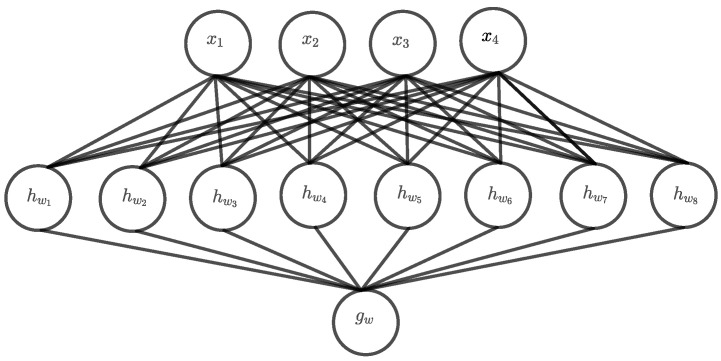
DNF architecture for parity n=4.

**Figure 3 entropy-24-01117-f003:**
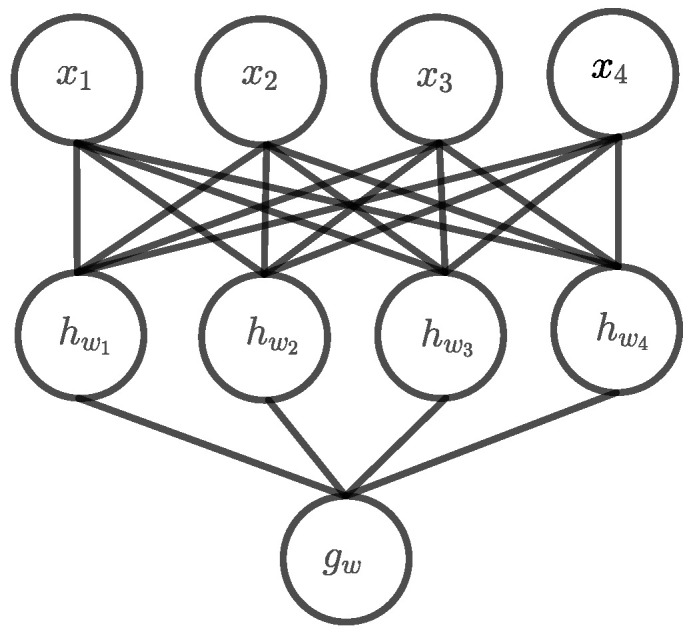
Exact architecture for parity n=4.

**Figure 4 entropy-24-01117-f004:**
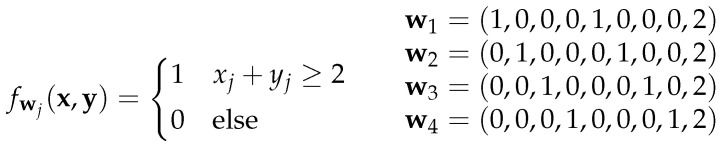
(**Left**) threshold function for the logical AND; (**Right**) example of hidden unit weights for n=4.

**Figure 5 entropy-24-01117-f005:**
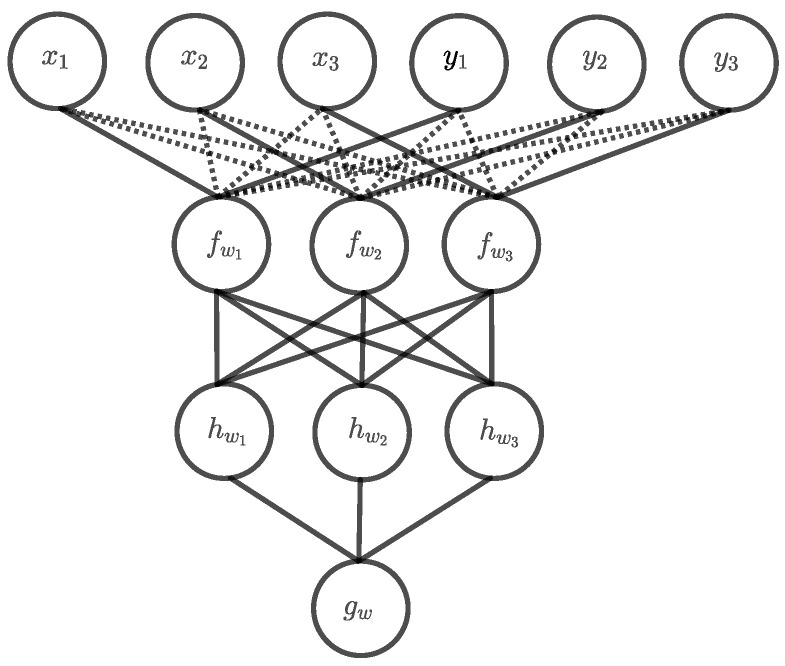
Deep exact architecture for Boolean inner product n=3.

**Figure 6 entropy-24-01117-f006:**
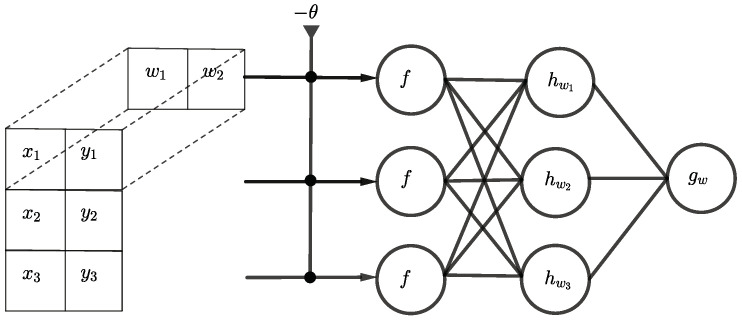
Deep partial architecture for the Boolean inner product n=3 using convolution.

**Figure 7 entropy-24-01117-f007:**
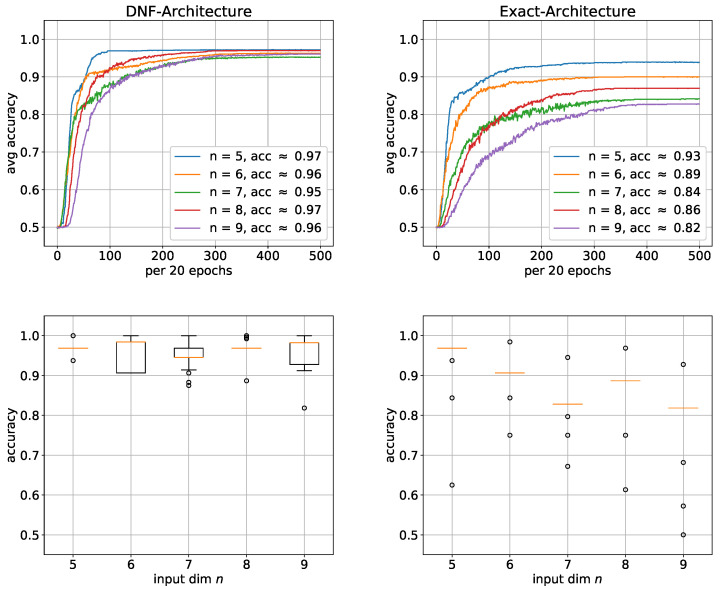
Learning the parity function with shallow architectures: average accuracies with corresponding boxplots below. (**Left**) DNF architecture; (**Right**) exact architecture. For each architecture, 100 trials were performed. The boxes represent the interquartile range (IRQ) beginning at the first quartile and ending at the third quartile. The line within the box is the median and the whiskers extend the boxes by 1.5 times the IRQ. Outliers are marked as circles.

**Figure 8 entropy-24-01117-f008:**
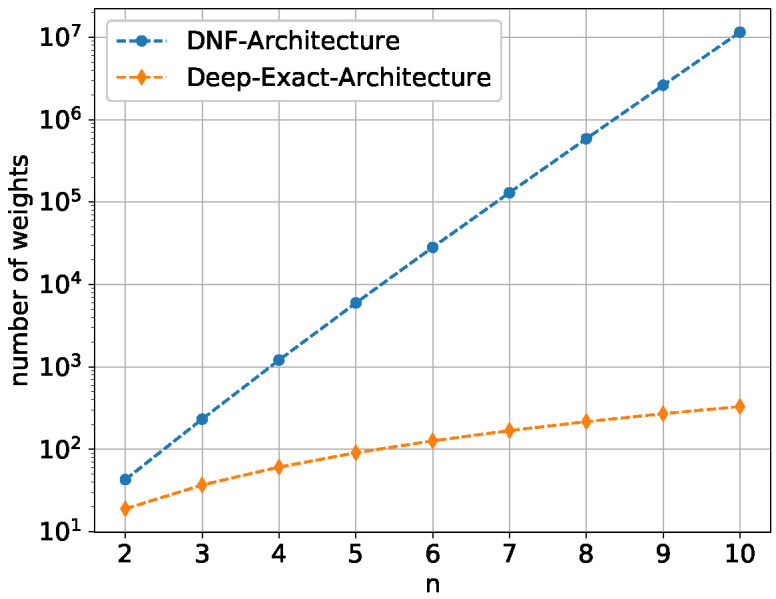
Number of weights for the DNF architecture and the deep exact architecture for n=2,3,…,10. The ordinate values are logarithmically scaled.

**Figure 9 entropy-24-01117-f009:**
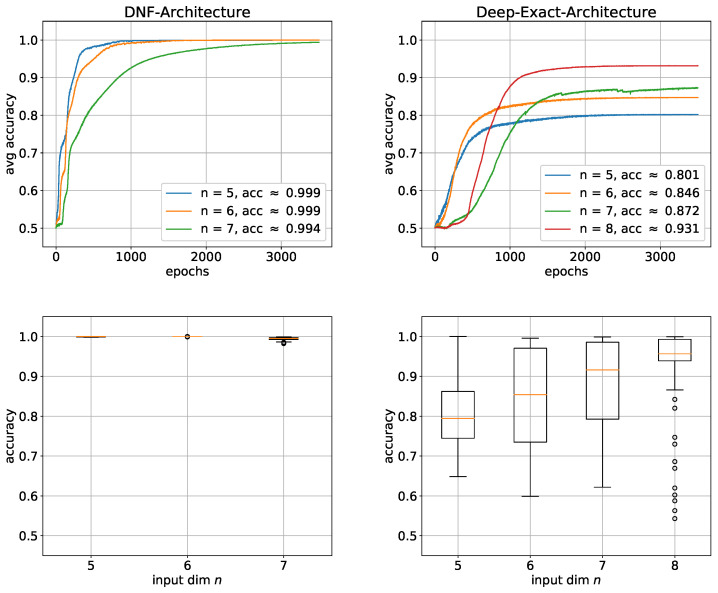
Learning the Boolean inner product with shallow and deep architectures: average accuracies with corresponding box plots below. (**Left**) DNF architecture; (**Right**) deep exact architecture. For each architecture, 100 trials were performed. The boxes represent the interquartile range (IRQ) beginning at the first quartile and ending at the third quartile. The line within the box is the median and the whiskers extend the boxes by 1.5 times the IRQ. Outliers are marked as circles.

**Figure 10 entropy-24-01117-f010:**
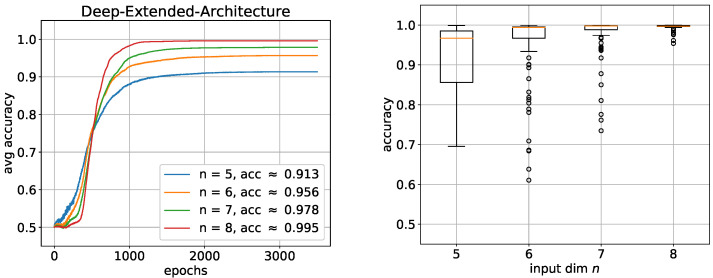
Learning the Boolean inner product with an additional two neurons in the first hidden layer of the deep exact architecture. (**Left**) average accuracies; (**Right**) corresponding box plots. For each architecture, 100 trials were performed. The boxes represent the interquartile range (IRQ) beginning at the first quartile and ending at the third quartile. The line within the box is the median and the whiskers extend the boxes by 1.5 times the IRQ. Outliers are marked as circles.

**Figure 11 entropy-24-01117-f011:**
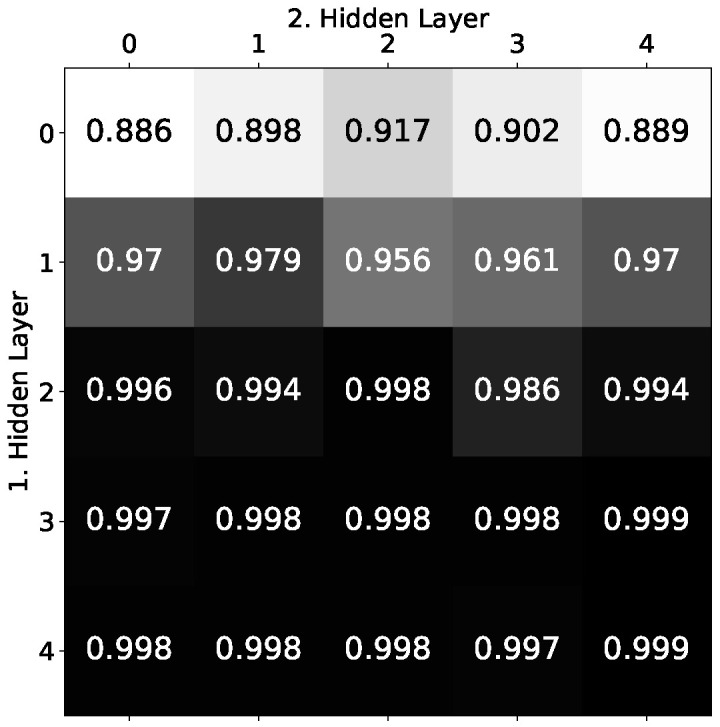
Matrix of average accuracies. Each element (a, b) of the matrix shows the average accuracy over 10 trials for architectures with *a* additional neurons in the first and *b* additional neurons in the second hidden layer.

**Figure 12 entropy-24-01117-f012:**
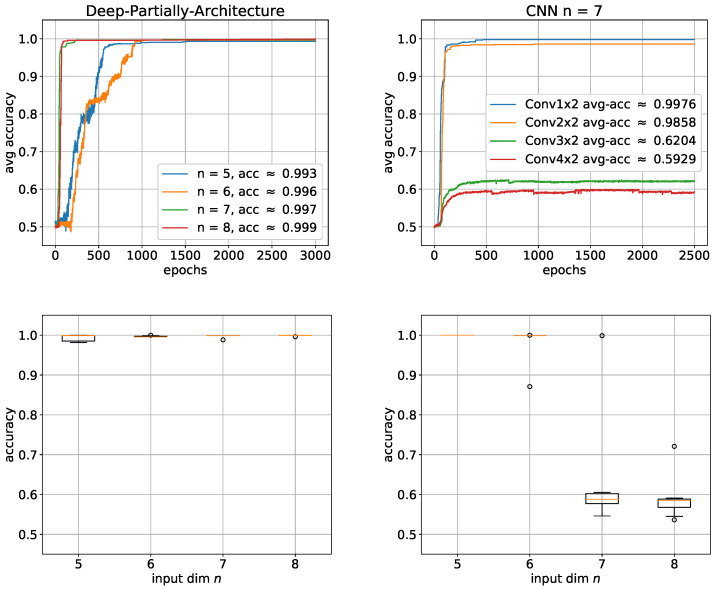
Learning the Boolean inner product with partially connected architecture: average accuracies with the corresponding box plots below. Left: deep partially architecture; Right: several filter sizes for n=7. For each architecture, 10 trials were performed. The boxes represent the interquartile range (IRQ) beginning at the first quartile and ending at the third quartile. The line within the box is the median and the whiskers extend the boxes by 1.5 times the IRQ. Outliers are marked as circles.

## Data Availability

Not applicable.

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
