# Peer review of "Learnability of the Boolean Innerproduct in Deep Neural Networks"

_entropy, 2022, doi:10.3390/e24081117_

Round 1
Reviewer 1 Report
This article presents an evaluation, named “Boolean Innerproduct” of the weights of a dense neural network, which is adapted to study the influence on performance in cases of different number of neurons and hidden layers for some specific tests. The change of performance is evaluated while densely connected neurons are replaced with convolutional kernels. The overall idea is not a certain novelty and it is expected to be exploited in a number of studies. However, this study misses an up-to-date research of the field and reference/comparison to similar methods.
The article contains substantial deficiencies in all sections: ‘Introduction (insufficient 15 references mainly before 2020) , Materials and Methods (missing definition of experiments, input data and performance metrics), Results (not properly explained and presented graphics and justification of the most important results), Discussion (missing comparison with other studies), Conclusions (not linked to specific results).
The article needs considerable changes following the detailed revision remarks. Please could you ensure that you provide a point-by-point response to each of my remarks and provide a document with any changes highlighted in any future draft of this article, if any.
Major revision remarks:
1. Title: “Deep neural networks” is quite general expression of the neural network architectures, but not limited to a shallow ONE, including dense (or fully connected) hidden layers whose boolean innerproduct is only under consideration of this study. The title should be reformulated to a more correct representation of the methods.
2. Abstract: Keep the journal guidelines for structured abstracts, including virtual sections Background, Methods, Results, Conclusions. I note the following inconsistencies in the abstract:
- Aims (or primary hypothesis) of the study are not properly defined.
- Missing information on the Materials: definition of the input data and planning of the experimental/simulation study.
- Missing numerical results.
- Conclusions NOT supported by the results.
3. The structure of the text is not following the journal’s recommendations for research articles, including major sections: 1. Introduction, 2. Materials and Methods, 3. Results, 4. Discussion. The article should be formatted accordingly with special attention to section “2. Materials and Methods”. Information about section 2.1 Materials is completely missing.
4. Section Introduction is insufficient. It refers only 15 references, while the number of articles using deep neural networks is progressively increasing in the last 2 years. The authors are required to considerably extend their Introduction. Furthermore, the list of References is ancient, including most studies before 2000. This is not representative to the state-of-the-art research and indicates insufficient knowledge in the field, including also unspecified information on similar (or close field) experiments as in this study.
5. Aims of the study are not properly defined in the last paragraph of Introduction.
6. Methods: The referenced methodological background [16-19] is of studies from 2012, 1989, 2001, 1969, which is too far behind current deep network technologies. It seems that the methods are not up to date and do not present interest for publication.
7. Page 2: “All fully connected architectures described here can be found in [? ].” -> Missing reference.
8. How do you consider the activation function of the hidden layer? How the results would change if different activation functions are used?
9. How do you consider the pooling and drop-out layers, which are typically connected in the neural architecture after the fully connected (or convolutional) layers?
10. How do you answer the common and most important question: “What is the importance of the input layer features?”
11. The term “Exact Architecture” is confusing. “Exact” is linked to the result but not to the “architecture”. Better use “convolutional layer architecture” or something that better describes the real architecture. Similar comment could be also linked to “disjunctive normal form (DNF) architecture”, which corresponds to dense neural architecture (but with untcommon name).
12. Section “2.1.1. Exact Architecture” -> the term “fitting exactly solution” is indefinite.
13. Section “2.2.1. Paritally Connected First Hidden Layer” -> The authors convert the task for 2D convolution while 1D convolution (commonly used for signal processing) would be quite mote comprehensive and simple example, so that 1D kernel closely corresponds to the 1D units in fully connected (dense) layers. Justify the reason for this complication.
14. Section Materials and Methods: Missing definition of the experiments/simulations and metrics for performance evaluation.
15. Section Materials and Methods: Missing definition of the input data.
16. Page 6: “Before we describe the results found …… is used or more precisely all possible examples will be used. As learning algorithm stochastic gradient descent (SGD) with Nestorov-momentum and an adaptive learning rate is applied.” -> The whole paragraph represents methodological settings and it is relevant to be presented in Methods. A lot of methodological details and data are, however, omitted so that the experiment is not reproducible.
17. Page 6: “The parity function is notorious for being very hard to learn with back-propagation and the results of our experiments confirm this expectation to some extent.” -> Justify this fact with appropriate references and information.
18. Figure 7: The content of the x-axis of the upper figure is unknown (the notation “per 20 epochs” is unknown). The caption “For each architecture 100 trials were performed.” -> Not corresponding to the results in the upper subplot. The curve seems as a training curve of one model.
19. Figure 7: The derivation and content of the bottom subplot is unclear.
20. Page 7: “Especially for the Exact-Architecture there is almost no variation within the sample of 100 trials.” -> This result is not correct. There is substantial variation of accuracy in the range 0.5-1 in Figure 7 (bottom).
21. Figure 8, the caption: “Number of weights … ” ->? Why the following formula is presented here but not as an equation in Methods. Methodological details should NOT be presented in figure’s caption (Results).
22. The results in Figure 7 and Figure 8 are NOT surprising but expectable. The networks architecture with less features is normally expected to be less accurate than the one with more features, which can be easily biased to represent the output.
23. Figures 7,9,10,12: Missing legend on the interpretation of the special symbols used in box-plots.
24. Figure 9. Input dimension n=8 is missing on the left.
25. Figures 10, 11 should be placed in section “3.2. Boolean Innerproduct” closely after their first call in the text.
26. Page 7: “Only the learning speed decreased with increasing n.” -> Indefinite notation. Undefined methods.
27. Page 7: “This could be an indication that the deep exact architecture needs a lot of training examples.” -> Confusing. Which are the training examples (more properly “samples”) in this study? Define in Methods.
28. Figure 12: The meaning of this figure is questionable, except that CNN has worse performance than Dense-NN for >2 CNN layers.
29. In overall, the results are not well justified in the main text. They are not enough illustrative for global conclusions.
30. Important conclusions are not drawn from the results.
31. Discussion: Compare your results and conclusions to other similar studies in recent years.
Author Response
Dear Reviewer,
we are really thankful for your time and afford. You can find the responses to your comments in the attached pdf. We hope that we could answer your question sufficiently.
Best regards,
Mehmet Erdal and Friedhelm Schwenker

Reviewer 2 Report
The authors of the paper suggest that one way to better understand deep learning architectures is to study logical functions. It is certainly true at the stage of improving one's own research skills in this topic. However, as they say themselves, further in-depth research is needed, and I encourage them to do so.
Currently, the subject matter presented in the work is discussed by many authors, that is why in my opinion the paper the topic fits in general the scope of the journal well. Nevertheless, there are some doubts regarding assumptions made to the proposed approach. Therefore, I would like ask authors to emphasize following aspects of the presented study:
a) Due to the lack of diligence in the preparation of the content of the manuscript, there are editorial errors that make it difficult or sometimes impossible to understand the essence of the proposed solution for example incoherent numbering of figures with the text description. This element requires careful review of the entire work and thorough improvement.
b) In the conclusions of the work, please provide more comprehensive explanations what is the real impact of the proposed solution a specially with references to other strategies/methods?
Author Response

(The authors gave the same response as above.)

Reviewer 3 Report
The topic is important but the research area is not new. The authors focus on the Boolean functions for analysing their impact on learning artificial neural networks.
After reading the paper I suggest correcting some parts of the paper.
State of the art should be a better present. Now it is part of the introduction and next chapters, but I suggest separating the chapter from the literature review. The literature review includes just 19 papers and just two present the papers are written in the last ten years. There is no new paper. I suggest making a literature review again and enlarging it. For example, the good papers in the area of ANN are PickupSimulo–Prototype of Intelligent Software to Support Warehouse Managers Decisions for Product Allocation Problem, Applied Science 2020; The most common type of disruption in the supply chain - evaluation based on the method using artificial neural networks, International Journal of Shipping and Transport Logistics 2021. What is more, the formatting of the literature is wrong (some lack the date of publication, lack of number or volume. The format of many positions is wrong.
There is a lack of the chapter Research methodology, so the structure of the paper is difficult to read. In research methodology, the aim should be clearly defined and the analysed case must be presented. It is not clear whether chapter 2 is part of the research or just a literature review. After that authors present the results. So the methodology was omitted.
Chapter 2 last paragraph finish with the lack of a link to literature: “… All fully connected architectures described here can be found in [? ].” It must be corrected.
As result the 10 trials (simulation cases) were presented, this number of trials is too low. General conclusions cannot base on just ten cases. I encourage authors to try more cases – the results could be different.
Author Response

(The authors gave the same response as above.)

Round 2
Reviewer 1 Report
The authors have slightly and insufficiently improved their article after the first revision round. There are still many questions which should be answered before the article gets some publishable form. The article needs essential revision of the English style of writing too.
Furthermore, I note the impolite tone while answering the reviewers! This intuitively makes reviewers more nagging and hungry for more questions.
Note that the authors didn’t prepared their template according to the journal guidelines and lines counts are missing, therefore I cannot link my questions to specific lines.
Important revision remarks:
- Substantial revision of the English style is necessary.
- Section “1.1. Circuit Complexity of the Boolean Inner Product” cannot be placed in Introduction! All methodological formulas should be disclosed and explained in section 2.1. “Mathematical Background” upon section 2. Methods.
- Introduction is normally published unsectioned. All prior research studies are described in a single consistent section. Section “1.2. Related Work” should be distributed as part of Introduction.
- Missing AIMS of the study in the final paragraph of Introduction! This is quite indefinite and insufficient information: “We will first give a detailed description and some backround informations about the architectures we studied in this work and then describe how we used SGD to train them.” Should be placed where expected and appropriate but not in the middle of Introduction.
- Section name “2. Methods and Materials” is out of standards. Any publication should strictly follow the format and guidelines of the journal linked to section name “2. Materials and Methods”, even though the authors judge themselves that “Materials” is not the first and important part of their work. Anyway, most authors describe their materials first.
- “The situation is quiet different in the Boolean subspace” -> incomprehensive statement.
- “is therefor defined” -> incomprehensive statement.
- “Fig. 8 shows a DNF-Architecture” -> Fig.8 placed after Fig.2??? Follow sequential numbering of figures. Figure 8 CANNOT be referred before Figs 3,4,5,6,7! If you need to refer some figure of Results just in Methods, this explicitly means that the text structure is WRONG! Such file numbering inconsistency is not publishable and could lead to rejection of the article.
- Figure 4, caption: Note that both notations on the right and left are reversed.
- Page 6: “For the Boolean inner product a 1D convolutional layer as input layer” -> Convolutional kernels cannot be an input layer but a hidden layer.
- “2.4. Experimental Setup” -> The title of the section is confusing. This is a simulation study but not an experimental study. Furthermore, the data used in the simulations are NOT described. This interrupts the understanding on the input feature map and the format of the output data. This setups the important question why binary cross-entropy is used as a loss function but not a weighted categorical cross-entropy, for example.
- Page 6: “sigmoid is used as activation function in all networks” -> explain where ‘sigmoid’ is used: in all convolutional layers or only in the final layer. The latter is a usual practice while other hidden layers use ‘relu’. Furthermore, explain why ‘softmax' activation was not used instead of ‘sigmoid’.
- Page 6: “LTU-network solution” -> The term LTU is not explained.
- Page 6: “As already explained in section 2 all possible examples are used as training data.” -> Clarify the statement. Data have not been explained. This is a big gap of misunderstanding.
- Page 7: “and the results of our experiments confirm this expectation to some extent.” -> Conclusion NOT supported by the results. Such conclusions cannot be made in the beginning of section Results, where the reader is still not acquaint with the results. Such statements should be justified with a direct reference to respective figures or tables.
- Page 7: 1st paragraph in section Results: The overall new (yellow) paragraph is NOT relevant in the beginning of Results. Turn an important note that the article got an absolutely unacceptable structure with a summary and conclusions before justification of any numerical results.
- Figures and tables should appear right after their first mention in the text. For example, figure 7 is not at its proper place. Correct other such inconsistencies.
- Figure 7: The term ‘acc’ in the legend as well as the y-axis label (avg accuracy) is not explained in Methods. Formulas in Methods should justify all performance metrics, which are calculated in results. The measurement units are also confusing. Normally performance is computed in percentages.
- Figures 7, 9,10: The legend or caption does not include any explanation about different marks in boxplots. The reader cannot interpret the content of these figures.
- “While DNF-Architecture grows exponentially with” -> I don’t see any exponential functions here! Furthermore, turn attention that the equations are not numbered.
- “Even for n=8, 100 trials with the DNF-Architecture would take several weeks” -> This is a speculation and indefinite statement. In fact, 100 trials in n=8 could take less than 3 days in my experience, depending on the machine.
- “So we performed experiments for the Deep-Exact-Architecture for n = 5, 6, 7, 8 and for the DNF-Architecture for n = 5, 6, 7.” -> How did you perform experiments? Misssing database description as required before
- Results, section “3.3. Partially Connected Network” -> It is confusing to see results for a network, which is not designed in methods. All methodological terms and network designs should be first introduced in methods.
- Page 10: “We again performed experiments for n = 5, 6, 7, 8, but this time only 10 trials per architecture in the expectation that the solution will be approximated much better, faster and more stable.” -> justify the reason to do this!
- Page 10: “again the result gets better and better as n increases” -> Poor style of writing scientific articles. There should be a difference between how you speak and how you write, no?
- Page 11: “varies very fast between different examples in the input space” -> Incomprehensive statement. Fast in terms of what?
- Page 11: “The hamming distance” -> Hamming distance
- Appendix A: make sure that you reference appendix A in a proper place. Otherwise, it is questionable why did you present this appendix.
- Author contributions: “please turn to the CRediT taxonomy for the term explanation. Authorship must be limited to those who have contributed substantially to the work reported.” -> This should not be part of the official text.
Author Response
Dear Editors ! Dear Reviewers!
Thank you again for your letter and for comments on our manuscript entitled“Learnability of the Boolean Innerproduct in Deep Neural Networks”.We thank the reviewers for their time and effort that they have put into reviewing the second version of the manuscript.Those comments are all valuable and very helpful for revising and improving our paper. We have studied comments carefully and have made corrections that are hopefully useful.
Thank you for allowing us to resubmit a revised version of the manuscript.
Please see attached PDF for the answer to your remarks.
Sincerely,
Mehmet Erdal and Friedhelm Schwenker.

Reviewer 3 Report
The paper was corrected regard to the comments and can be published in this form.
Author Response
Dear Editors ! Dear Reviewers!
Thank you again for your letter and for comments on our manuscript entitled“Learnability of the Boolean Innerproduct in Deep Neural Networks”.We thank the reviewers for their time and effort that they have put into reviewing the second version of the manuscript.Those comments are all valuable and very helpful for revising and improving our paper. We have studied comments carefully and have made corrections that are hopefully useful.
Thank you for allowing us to resubmit a revised version of the manuscript.
Sincerely,
Mehmet Erdal and Friedhelm Schwenker.
Round 3
Reviewer 1 Report
The authors have improved the comprehension of their paper. I have minor revision comments:
1. AIMS not written properly. They should be defined after related studies and just based on the differences with them, highlighting the novelty of this study.
2. Figures 7, 9, 10, 12 need a legend. The red lines, box-plots, whisker plots and dots can mean anything, e.g. mean valie, standard deviation, min-max range or median value, some percentage range around median value (frequently interquartile but not always), non-outlier range. Sometimes dots can mean outliers, in other case they can mean extremes. So, the interpretation of any statistical box-plot needs explanation (if you cannot graphically provide such, then extend the caption). But you NEED to do this!
3. All figures should be referenced in the text as Figure but NOT Fig.
4. (see A for a proof), -> means nothing. The reference to Appendix A should be clearly identified.
Author Response
Dear Reviewer,
we thank you very much again for your valuable feedback and your time. We hope that we could further improve our paper based on your comments.
Here are our answers:
1. AIMS not written properly. They should be defined after related studies and just based on the differences with them, highlighting the novelty of this study.
Answer
We dedicated an own subsection "2.2 Learning of the Boolean Inner Product" to state our aims in more detail.
2. Figures 7, 9, 10, 12 need a legend. The red lines, box-plots, whisker plots and dots can mean anything, e.g. mean valie, standard deviation, min-max range or median value, some percentage range around median value (frequently interquartile but not always), non-outlier range. Sometimes dots can mean outliers, in other case they can mean extremes. So, the interpretation of any statistical box-plot needs explanation (if you cannot graphically provide such, then extend the caption). But you NEED to do this!
Answer
We added an explanation for the boxplots in the captions of all figures witch boxplots.
3. All figures should be referenced in the text as Figure but NOT Fig.
Answer
All figures are referenced with "Figure" in text now.
4. (see A for a proof), -> means nothing. The reference to Appendix A should be clearly identified.
Answer
The appendix is now referenced with an additional clear explanation.
Sincerly,
Mehmet Erdal